**Data Availability Statement:** All relevant data are within the paper and its Supporting Information.

**Funding:** This study was funded by BASF Corporation under the grant number 4-20-B00454-

# White LED intensities during co-cultivation affect the *Agrobacterium*-mediated soybean (*Glycine max*) transformation using mature half seeds as explants

Xiaonan Shi[1], Baochun Li[2], Marcela Rojas-Pierce[3], Ricardo Hernández 🆔[1]*

**1** Department of Horticultural Science, North Carolina State University, Raleigh, NC, United States of America, **2** BASF Agricultural Solutions, Durham, NC, United States of America, **3** Department of Plant and Microbial Biology, North Carolina State University, Raleigh, NC, United States of America

* rhernan4@ncsu.edu

## Abstract

The transition of light fixture from fluorescent light to light-emitting diodes (LEDs) in growth chambers prompts a reevaluation of current practices in plant biotechnology. *Agrobacterium*-mediated transformation is crucial for genetic engineering and genome editing in soybean (*Glycine max*). The critical co-cultivation step of soybean transformation occurs under light condition. Current protocols for co-cultivation in soybean transformation lack a standard for light intensity. In the present study, the objective is to investigate the effect of light intensity during co-cultivation on soybean transformation efficiency. Five light intensities were implemented during five days of co-cultivation: 50, 100, 150, 190 $\mu mol \cdot m^{-2} \cdot s^{-1}$ of white LEDs in addition to 100 $\mu mol \cdot m^{-2} \cdot s^{-1}$ of fluorescent light. After co-cultivation, all the explants underwent shoot induction and elongation with selection pressure, rooting and acclimation under uniform condition. The experiment was conducted with two selectable markers, *hppdPf-4Pa* and *bar*, separately, investigating whether the potential light effects vary due to the marker-associated pathways. The positive PCR analysis of rooted *in vitro* plants suggested successful transformation events achieved under both selectable markers across all light treatments ranging from 2.4% to 6.9%. Increasing LED light intensity during co-cultivation resulted in different transformation efficiencies between the two selectable markers. Results indicated that increasing the light intensity during co-cultivation led to a linear increase in transformation efficiency when shoot regeneration was under 4-Hydroxyphenyl-pyruvate dioxygenase (HPPD) inhibitor selection. No difference in transformation efficiency was detected among the treatments under glufosinate selection. Furthermore, when selection occurred with HPPD inhibitor, variation of transformation efficiency was also observed between fluorescent light and white LED at 100 $\mu mol \cdot m^{-2} \cdot s^{-1}$. The results highlight the significance and potential applications of investigating the impact of light on transformation efficiency.

3. Ricardo Hernandez (RH) received this award. The study was supported by BASF Agricultural Solutions, located at 2 TW Alexander Drive, Durham, North Carolina 27709, Research Triangle Park, USA. Baochun Li (BL), an employee of BASF Agricultural Solutions, received salary and other funding from the company. More information about BASF can be found at BASF Agricultural Solutions' website (https://agriculture.basf.us/). Dr. Baochun Li, as an employee of BASF and co-author, contributed to the study design alongside authors at the University. Dr. Li provided valuable contributions to data interpretation. Dr. Li and BASF Corporation supported the decision to publish and the submission of the manuscript. Dr. Li also reviewed the manuscript and offered insights.

**Competing interests:** The authors have declared that no competing interests exist.

## Introduction

The rapid emergence of light-emitting diodes (LEDs) technology has revolutionized lighting systems, offering high energy efficacy and prolonged durability [1]. Consequently, there has been a notable transition from fluorescent to LED lights in various industrial sectors. In plant biotechnology, growth chambers have been increasingly outfitted with LED fixtures [2]. LED fixtures have 200% higher fixture efficacy (2.2–3.2 µmol J$^{-1}$) compared to fluorescent lamps (~0.84–0.95 µmol J$^{-1}$) [1] which are commonly used on tissue culture chambers. Therefore, with LEDs, it is more feasible to increase light intensity with the same power input. Previous studies have demonstrated enhanced plant growth with increased light intensity and variations in endogenous phytohormone levels for soybean plants *ex vitro* under different light intensities [3, 4]. Additionally, while white LEDs produce a broad spectrum like fluorescent lamps, there are differences in the light spectrum profiles between the two. Even though several studies have demonstrated successful explant growth *in vitro* under various light intensities of LEDs, question remains whether different response of explant growth and development exist between fluorescent and LEDs. It is imperative to evaluate plant tissue responses to LEDs compared to fluorescent lights to ensure the successful adoption of LED technology and uncover new light applications in plant science, particularly in plant biotechnology. This study includes fluorescent light as industry standard to compare with LEDs treatments and provide insights for light source transition for soybean transformation process.

Soybean is a major agronomic crop with significant economic and food security value. The soybean genetic transformation is critical for crop improvement by genetic engineering and genome editing [5]. Various explant types have been employed in *Agrobacterium*-mediated transformation systems, including cotyledonary node from immature seed [6], hypocotyl from germinated seedlings [7], and half-seed from mature seed [8]. The use of half-seed explants offers significant advantages over methods involving immature seeds and seedlings, as these latter approaches require prolonged plant care and maintenance in greenhouses or growth chambers prior to the transformation process. From which half-seed explants derive, mature seeds are easy to store and avoid the need for greenhouse facilities. Additionally, the half-seed method enables rapid and direct shoot organogenesis from the meristem, bypassing the callogenesis stage. This is crucial for expediting the regeneration process of genetically modified shoots. Notably, the meristem on half-seeds is exposed to *Agrobacterium* during infection and co-cultivation and serves as the site for shoot induction. Co-cultivation of *Agrobacterium* and infected explants constitutes a crucial step in successful transformation, during which foreign DNA integrates into the plant genome [9]. In soybean transformation, the co-cultivation stage occurs under light conditions. Meristem activity intricately links to light signaling pathways involving phytohormones such as cytokinins and auxins [10] and the direct regulation of gene expression for cell proliferation [11]. Previous studies have explored the effects of light signaling during co-cultivation on transformation efficiency, focusing on light absence or presence in tobacco (*Nicotiana benthamiana*) [12] and tepary bean (*Phaseolus acutifolius*) [13]. However, the impact of light intensity during co-cultivation on transformation efficiency remains largely unexplored regardless of the plant species and the implemented light source. Moreover, a wide range of light intensities, ranging from 50 µmol·m$^{-2}$·s$^{-1}$ to 140 µmol·m$^{-2}$·s$^{-1}$, has been utilized across previous soybean transformation studies [14–16]. This fair-sized variation of light intensity could bear potential influence on inconsistent transformation efficiency across the studies. The present study initiates efforts of evaluating the effects of light intensity during the co-cultivation stage of soybean transformation.

The *bar* gene is a commonly used selective marker in soybean transformation, with glufosinate (an herbicide) typically applied as the selective agent [17]. The bar gene encodes a

phosphinothricin acetyltransferase (PAT) [18]. PAT can detoxify phosphinothricin, commercially known as glufosinate, which inhibits glutamine synthetase [19]. However, challenges arise in this process, including chimeric plants and escaping transgenic events [20]. The *hppd* gene, integral to the photosynthesis pathway, holds potential as an alternative selective marker for plant transformation [21]. One of *hppd* genes is *hppd-Pf-4Pa*, derived from *Pseudomonas fluorescens* (ISAAA database). Transformed soybean plants containing *hppd-Pf-4Pa* demonstrate resistance to HPPD inhibitors (herbicides), while wild-type soybean plants exhibit "bleaching" effects (white leaves) in the presence of HPPD inhibitors [21]. Leveraging *hppd-Pf-4Pa* as a selective marker provides the advantage of a distinguishable phenotype during selection, dueto the contrasting color difference between plant tissues with and without the gene. Moreover, employing *hppd-Pf-4Pa* may mitigate the chimeric plants resulting from transformation. However, the implementation of HPPD inhibitors as selective agents in soybean transformation has yet to be fully established in the publications, regarding the applied selection pressure. This current research explores a selection protocol utilizing tembotrione, a member of the HPPD inhibitors family, as the selective agent in the *Agrobacterium*-mediated transformation process of soybean. Outcome of selection pressure by tembotrione can set in motion for its application in transformation of many plant species. Given the difference between the pathways of the *bar* gene and *hppd-Pf-4Pa*, investigation result indicates whether the choice of selective marker impacts potential light effects on transformation efficiency.

The objective of the present study was to determine the effects of white LEDs intensity during co-cultivation on soybean transformation using *bar* and *hppd-Pf-4Pa* as selective markers. The study hypothesized that an increase in light intensity during co-cultivation improves the efficiency of *Agrobacterium*-mediated transformation in soybeans, but the effects are dependent on the type of selective marker used in the shoot organogenesis phase. Furthermore, the present study confirmed the possibility of using HPPD inhibitors as selective agents in the transformation process.

## Materials and methods

### Explant preparation

The experiments used soybean cultivar 'Thorne' as the plant material. The seeds were provided by the BASF Innovation Center. Half-seed preparation was done as previously reported [15]. A single layer of mature soybean seeds was placed on petri dishes and subjected to surface sterilization using chlorine gas, specifically a mixture of 100 ml NaClO and 3.5 ml HCl (12N). This process took place in a large Pyrex desiccator for 16 hours. After sterilization, the petri dishes containing the soybean seeds were uncovered in a laminar flow hood for at least 30 minutes to allow excess chlorine gas to dissipate. The sterile seeds were then rinsed with sterile distilled water and imbided in 250 ml flasks filled with sterile distilled water in the dark at 24°C for 16 hours. The seeds were loaded up to the 100 ml scale line. The imbibed seeds were prepared as half-seeds in a laminar hood by trimming the embryonic axis of the soybean seed to approximately one-third of its length, removing the radicle and part of the hypocotyl. The seed was then split longitudinally into two halves at the hilum using a #10 blade. If any seed coat remained on the half-seed, it was removed. The plumule, which refers to the first true leaves of the soybean seed, was removed from the cotyledon to expose the shoot meristem on the half-seed.

### *Agrobacterium* preparation, infection, and co-cultivation

*Agrobacterium tumefaciens* strain EHA105 carrying the plasmid (pBNCL2020) construct depicted in Fig 1 was used for HPPD inhibitor and glufosinate selection, which was provided

**Fig 1. Schematic map of T-DNA that contains *hppdPf-4Pa* and *bar* and neighboring region for *Agrobacterium* selection.** The T-DNA consists of border sequences (LB and RB), promoter PcsvmvXYYZ that drives *hppdPf-4Pa* fused with coding sequence of an optimized transit peptide derivative *TPotpY-1Pf*, promoter P35S3 that drives *bar*. The neighboring region includes an antibiotic-resistant site *aadA* that is driven by promoter PaadA.

by BASF Innovation Center (Ghent, Belgium). The *A. tumefaciens* was streaked on YEP media consisting of 10 g/L yeast extract, 5 g/L NaCl, 10 g/L peptone, and 15 g/L agar, with a pH of 7.0. This media was supplemented with spectinomycin (100 μg/ml) and streptomycin (100 μg/ml). After an incubation period of 48 hours at 28°C, a single colony was isolated. The isolated colony was then cultured in 3 ml of starter liquid YEP media, which contained 10 g/L yeast extract, 5 g/L NaCl, and 10 g/L peptone, with spectinomycin (100 μg/ml) and streptomycin (100 μg/ml). The starter liquid was placed on a shaker at 250 rpm and 28°C overnight. Subsequently, 100 μl of the starter media was transferred to 100 ml of liquid YEP media in a 250 ml flask, again supplemented with spectinomycin (100 μg/ml) and streptomycin (100 μg/ml). The *Agrobacterium* culture was then incubated on a shaker at 250 rpm and 28°C until the $OD_{600}$ reading reached a range of 0.6 to 0.8. The bacterial pellet was collected by centrifugation at 4000 rpm and 20°C for 15 minutes. This pellet was then suspended in infection medium, which consisted of B5 salt, B5 vitamins, 3.9 g/L MES, 0.25 mg/L gibberellic acid ($GA_3$), 1.67 mg/L 6-Benzylaminopurine (BAP), 30 g/L sucrose, 40 mg/L acetosyringone, with a pH of 5.4, at an $OD_{600}$ range of 0.6 to 0.8.

The prepared half-seeds were submerged in the infection media containing *Agrobacterium* for a duration of 40 minutes. After the incubation, the half-seeds were drained and transferred to the co-cultivation phase. Placed adaxial side up in petri dishes, the half-seeds were positioned on three layers of filter papers soaked with co-cultivation media. The co-cultivation media consisted of B5 salt, B5 vitamins, 3.9 g/L MES, 0.25 mg/L GA), 1.67 mg/L BAP, 30 g/L sucrose, 40 mg/L acetosyringone, 400 mg/L L-cysteine, and 154.2 mg/L dithiothreitol, with a pH of 5.4. The co-cultivation process lasted for five days under various treatments of light intensity. Each petri dish contained 20 half-seeds.

## Light treatments during co-cultivation

Five light treatments were implemented during co-cultivation phase, including four light intensities of white LEDs (TS 600, 100 W, Mars Hydro, Shenzhen, Guangdong, China): 50, 100, 150, 190 μmol·m$^{-2}$·s$^{-1}$, and one light intensity of fluorescent light (T5s, 4100 Kelvin, 24W, Philips, Amsterdam, Netherlands): 100 μmol·m$^{-2}$·s$^{-1}$, which is a common fixture in tissue culture chambers. The light composition is shown in Fig 2 and Table 1, which was measured by a spectroradiometer (PS-200, Apogee Instruments, Inc. Logan, UT, USA). All co-cultivation took place in a walk-in growth chamber that had separate compartments. A light intensity map was created in each compartment, specifically at each Petri dish position targeting the desired light intensity using white LED and fluorescent fixtures. Each treatment consisted of four to five petri dishes, resulting in a total of 80 to 90 explants for each treatment. This experiment was conducted six times. The temperature on the surface of each petri dish was monitored using thermocouples (0.005-gauge, T-type, Omega Inc., Stamford, CT, USA). The temperature, $CO_2$ concentration, and relative humidity within the growth chamber were measured and recorded using a data logger (CR1000, Campbell Scientific Inc., Logan, UT, USA). All data were collected every five seconds and recorded as averages every five minutes

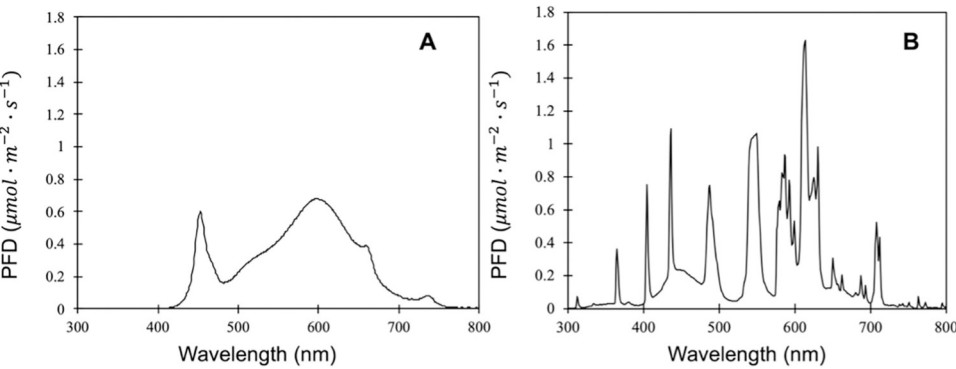

**Fig 2.** Light spectrum of white LED (A) and fluorescent (B) used in the present study.

throughout the duration of the experiment. The environmental data is summarized in Table 2. Co-cultivation period was five days, suggested by previously published work of *Agrobacterium*-mediated transformation using soybean half-seeds [11, 14, 15].

## Shoot induction and elongation

After five days of co-cultivation, all half-seeds from five light treatments were moved to the shoot regeneration process, involving shoot induction and shoot elongation phases. The basic shoot induction medium (SIM) is composed of full strength of B5 salts, B5 vitamins, 0.59 g/L MES, 1.67 mg/L BAP, 30 g/L sucrose, 7 g/L agar, 100 mg/L timentin, 200 mg/L cefotaxime, 50 mg/L vancomycin, pH 5.7. For glufosinate selection, SIM included 6 mg/L of glufosinate. Tembotrione was used as the HPPD inhibitor for selection in this experiment. Various rates of tembotrione were evaluated for its "bleaching" effects on shoot regeneration of soybean (Fig 3). Tembotrione selection concentration was determined at $0.15$ -$\mu mol \cdot L^{-1}$ in SIM. After two weeks of incubation in SIM, soybean explants with green shoots were selected. The induced leaves of chosen soybean explants were trimmed off, and the cotyledons with remaining shoot meristem were transferred to fresh SIM for two weeks.

After four weeks of shoot induction, the shoot clusters with green shoots were isolated and transferred to shoot elongation medium (SEM). SEM consisted of full strength of MS salts, B5 vitamins, 0.59 g/L MES, 50 mg/L L-asparagine, 100 mg/L L-pyroglutamic acid, 0.1 mg/L Indole-3-acetic acid (IAA), 0.5 mg/L $GA_3$, 1 mg/L zeatin-R, 30 g/L sucrose, 7 g/L agar, 100 mg/L timentin, 200 mg/L cefotaxime, 50 mg/L vancomycin, pH 5.7. The chemical selections were again used at $0.15$ $\mu mol \cdot L^{-1}$ tembotrione and 6 mg/L glufosinate. The soybean shoots

**Table 1. Light spectrum composition of white LED and fluorescent.** Photon flux density of Photosynthetically Active Radiation (PAR: 400–700 nm) and Extended Photosynthetically Active Radiation (ePAR; 400–750 nm) are present in $\mu mol \cdot m^{-2} \cdot s^{-1}$. Light spectrum of UV (300–400 nm), blue (400–500 nm), green (500–600 nm), red (600–700 nm), and far-red (700–800 nm) are present in percentage of total photon flux (300–800 nm). R/FR is the photon flux density ratio of red to far-red light. PSS represents Phytochrome Photostationary State. The light composition of white LED and fluorescent light was measured by a spectroradiometer.

| Light | PAR[*] | ePAR[**] | UV | Blue | Green | Red | Far-red | R/FR | PSS[***] |
|---|---|---|---|---|---|---|---|---|---|
| | $\mu mol \cdot m^{-2} \cdot s^{-1}$ | | % | | | | | | |
| White LED | 102 | 105 | 0.05 | 17.6 | 42.6 | 36.4 | 3.35 | 10.86 | 0.85 |
| Fluorescent | 99 | 104 | 3.06 | 24.66 | 34.88 | 32.63 | 4.77 | 6.84 | 0.83 |

[*]PAR: Photosynthetically Active Radiation, 400–700 nm

[**]ePAR: Extended Photosynthetically Active Radiation, 400–750 nm

[***]PSS: Phytochrome Photostationary State, the ratio of far-red phytochrome $P_{fr}$ to total phytochrome $P_{total}$.[22]

**Table 2. Environmental data during co-cultivation for HPPD inhibitor selection and glufosinate selection.** Values are present as mean ± standard deviation (n = 6).

| Selection Marker | Light Source | Targeted Light Intensity ($\mu mol \cdot m^{-2} \cdot s^{-1}$) | Actual Light Intensity* ($\mu mol \cdot m^{-2} \cdot s^{-1}$) | Temperature** (°C) | Photoperiod (hour) | $CO_2$*** ($\mu mol/mol$) | Relative Humidity*** (%) |
|---|---|---|---|---|---|---|---|
| HPPD inhibitor | White LED | 50 | 51.2 ± 0.9 | 24.6 ± 0.1 | 16 | 554.4 ± 6.8 | 60.2 ± 9.3 |
| | | 100 | 100.6 ± 0.7 | 24.9 ± 0.1 | | | |
| | | 150 | 151.4 ± 1.4 | 24.7 ± 0.3 | | | |
| | | 190 | 190.1 ± 1.2 | 25.3 ± 0.6 | | | |
| | Fluorescent | 100 | 100.1 ± 1.2 | 24.4 ± 0.2 | | | |
| Glufosinate | White LED | 50 | 51.0 ± 0.6 | 24.7 ± 0.1 | 16 | 591.7 ± 7.3 | 64.6 ± 5.0 |
| | | 100 | 101.6 ± 0.7 | 25.0 ± 0.1 | | | |
| | | 150 | 150.4 ± 1.1 | 24.9 ± 0.1 | | | |
| | | 190 | 190.5 ± 1.5 | 25.2 ± 0.3 | | | |
| | Fluorescent | 100 | 100.3 ± 1.0 | 24.6 ± 0.2 | | | |

*Light intensity was measured with *LI-COR* LI-190 quantum sensor at 5 cm above the platform.

**Temperature was measured with thermocouple right above the petri dish lid per treatment and recorded every five minutes.

***$CO_2$ and relative humidity of the chamber were measured by sensors and recorded every five minutes.

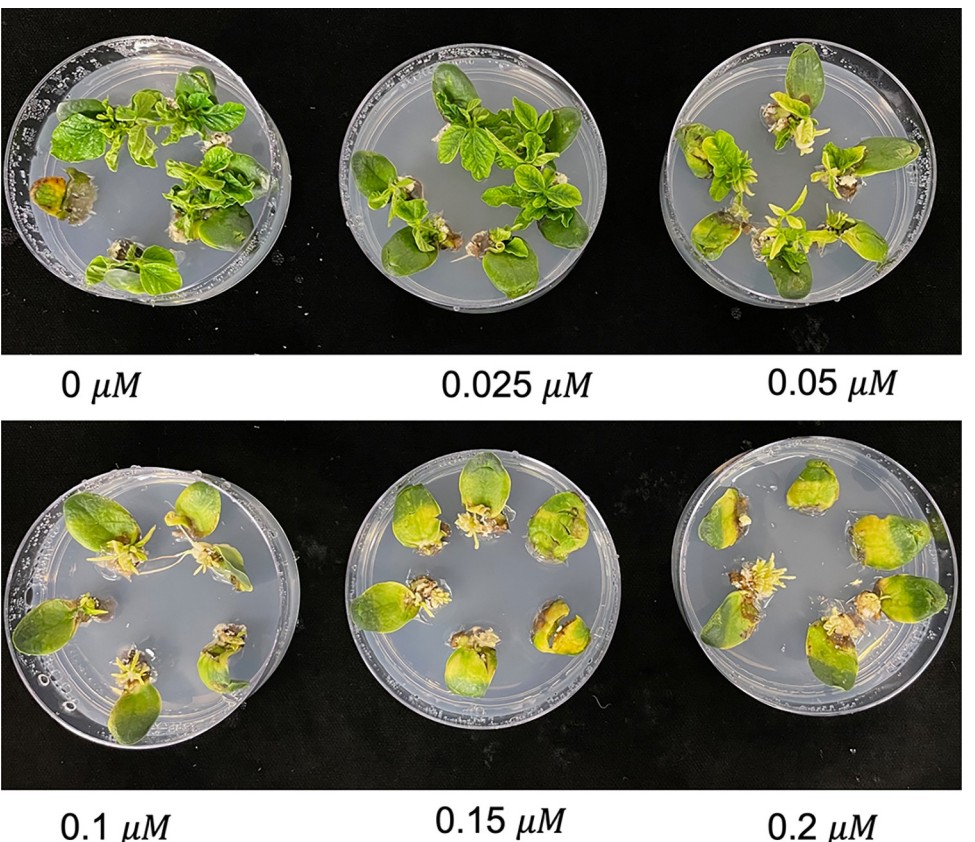

**Fig 3. The "bleaching" effects of various rates of tembotrione on soybean shoot regeneration.** The shoots present in the figure were 28-day-old culture in shoot induction medium.

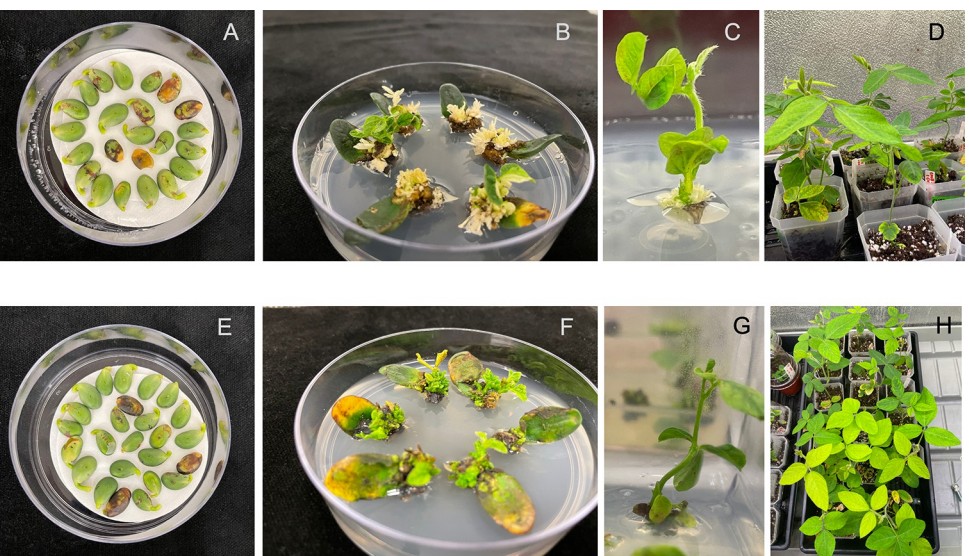

**Fig 4. Regeneration process of transgenic soybean plant.** The top panel is HPPD inhibitor selection. The bottom panel is glufosinate selection. Soybean half-seed explants after five days of co-cultivation (A and E). Explants in shoot induction medium (B and F). Elongated soybean shoots in shoot elongation medium (C and G). Potential transgenic soybean plants under acclimation (D and H).

were sub-cultured every two weeks until the green shoots elongated to higher than 3 cm. The elongated shoots were harvested and transferred to rooting medium. The shoot regeneration process is shown in Fig 4 (B&C: under tembotrione selection; F&G: under glufosinate selection).

## Rooting and acclimation

Rooting medium contained ½ strength of MS salts, ½ B5 vitamins, 15 g/L sucrose, 7 g/L agar, pH 5.6. The cut end of the soybean shoot was dipped into 1 mg/L Indole-3-butyric acid (IBA) for one minute prior to being inserted to rooting medium. Once the roots were formed, the soybean plants were moved to the acclimation stage for further establishment.

The shoot regeneration, rooting, and acclimation process took place in a walk-in chamber, where the environmental condition was monitored and recorded by a data logger (UX100-003, Onset®, HOBO Data Logger, Bourne, MA, USA). The condition was maintained at temperature of $26 \pm 0.7°C$, white LED (Arize® Life, BRV full spectrum, 32W, GE Current Lighting Solutions, Greenville, SC, USA) light intensity of $100 \pm 3.3$ µmol·m$^{-2}$·s$^{-1}$, and photoperiod of 18 hours, relative humidity of $39 \pm 6.9\%$ for all treatments.

## Transgene confirmation

The genomic DNA of the youngest mature leaf on acclimated soybean plants was extracted using adapted method from Edwards et al. 1991 [23]. Then the genomic DNA was evaluated for the transgene presence by PCR amplification of two amplicons for each selectable marker. For the amplification of *hppdPf-4Pa*, a pair of specific primers was designed. The forward primer "hppd-F" (GGATGGTGTATTGGGCTAACT) was designed to target the 5' region of *hppdPf-4Pa*, while the reverse primer "hppd-R" (CTGGTGCTGACATAGCTTTAGA) was designed to target the 3' region of *hppdPf-4Pa*. To amplify *bar* gene, the forward primer "bar-F" (AAGTCCAGCTGCCAGAAA) and reverse primer "bar-R" (CGAGGCGCTCGGATATG) were

designed to target 5' and 3' region of *bar*, respectively. The amplified PCR product was then visualized through gel electrophoresis (2% w/v agarose) to confirm the transgene presence.

## Transformation efficiency determination

Based on the PCR analysis for selective maker presence in the leaf tissue, the number of soybean plants carrying transgene was counted. Transformation efficiency (%) was calculated as Eq (1) for every repetition (co-cultivation) of each selective marker.

$$\frac{Number\ of\ plants\ with\ transgene\ confirmed\ by\ PCR}{Number\ of\ explants\ in\ co-cultivation} \times 100 \tag{1}$$

## Statistical analysis

The present study used a completely randomized design. Each soybean half-seed was randomly assigned to a treatment. The experiment was repeated six times. Each repetition contained 70–80 half-seed explants per treatment. The transformation efficiency of HPPD inhibitor selection was analyzed by simple linear regression model across increasing light intensity during co-cultivation. The transformation efficiency of glufosinate selection was analyzed by ANOVA analysis. All the data analysis was conducted in JMP software (JMP Statistical Discovery LLC, Cary, NC, USA).

## Results

### HPPD inhibitor selection

The transformation was confirmed by PCR analysis. Under tembotrione selection, the transformation efficiency responded linearly to increasing light intensity of white LED during co-cultivation (Fig 5). For example, soybean half-seeds that were co-cultured at 190 $\mu$mol·m$^{-2}$·s$^{-1}$ had 2.2 times (116%) greater transformation efficiency than those co-cultured at the lowest light intensity (50 $\mu$mol·m$^{-2}$·s$^{-1}$) of white LED. The 190 $\mu$mol·m$^{-2}$·s$^{-1}$ of white LED during co-cultivation yielded the highest transformation efficiency of 5.2%. Meanwhile, different mean value of the transformation efficiency was observed (not statistically different) between white LED and fluorescent light at 100 $\mu$mol·m$^{-2}$·s$^{-1}$. The transformation efficiency was 5.4% from the co-cultivation at 100 $\mu$mol·m$^{-2}$·s$^{-1}$ of fluorescent light, which was similar to the result from the co-cultivation at 190 $\mu$mol·m$^{-2}$·s$^{-1}$ of white LED.

### Glufosinate selection

The transformation efficiency was determined by using Eq (1) after confirming transformation event by PCR analysis on rooted soybean plants. Under glufosinate selection, the transformation efficiency showed no significant difference when co-cultivation took place under various light intensities (Table 3). Neither did the transformation under fluorescent light outperform the white LED at 100 $\mu$mol·m$^{-2}$·s$^{-1}$ or any other intensity. The transformation efficiency varied from 2.8% to 6.9% with big variation among the treatments.

## Discussion

This study revealed a linear increase of transformation efficiency under HPPD inhibitor selection, but not under glufosinate selection, when increasing light intensity during co-cultivation in soybean. This outcome indicates that light intensity during co-cultivation affects transformation efficiency possibly depending on the transgene and its expression. Prior to the present study, we explored the effect of the same range of light intensities during co-cultivation on

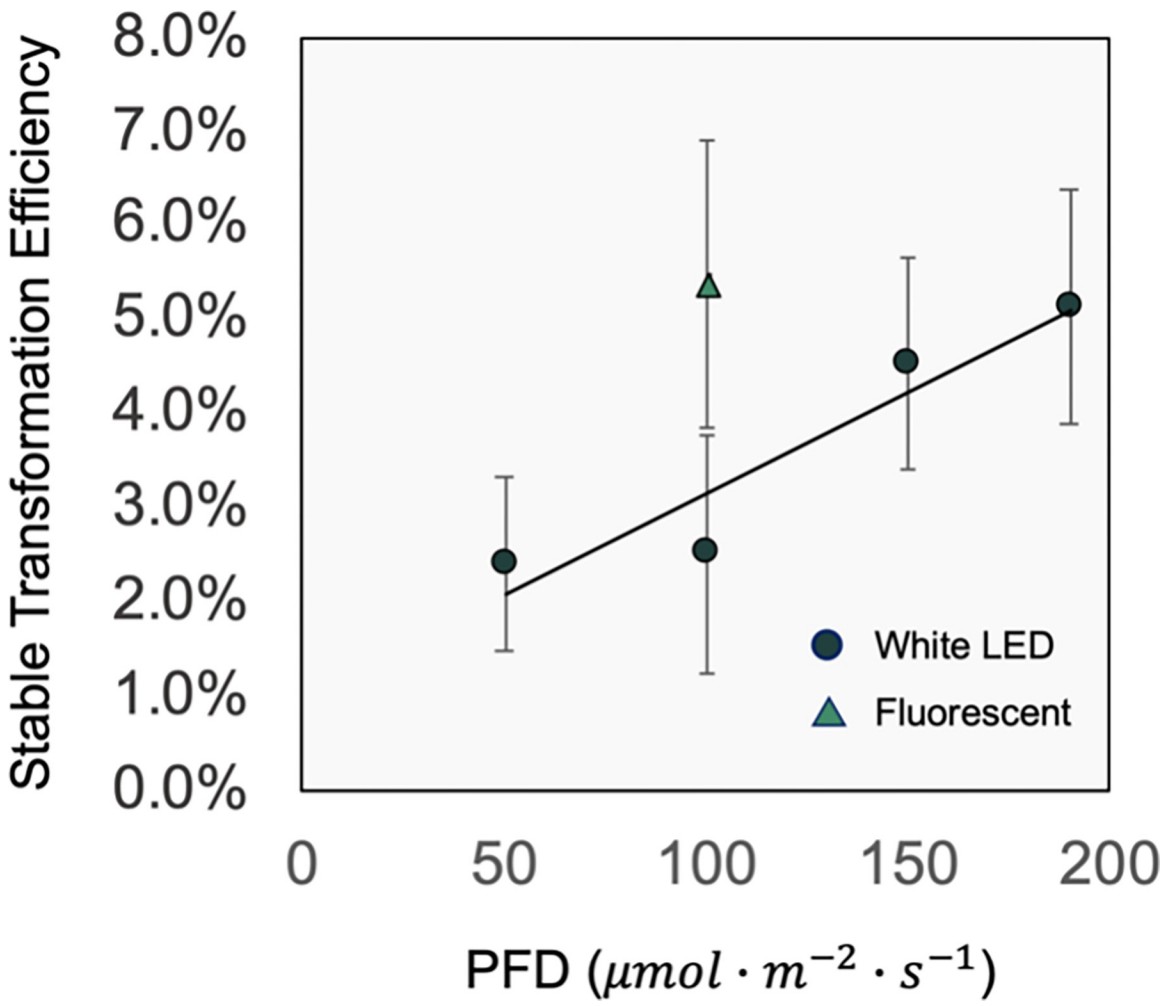

**Fig 5. Transformation efficiency response to light intensity during co-cultivation under tembotrione selection.** The simple linear regression model was significant with p-value of 0.0499, y = 0.022x + 1.0. PFD: photon flux density. The circle symbols represent the values from white LED treatments. The triangle symbol represents the value from fluorescent light. Standard errors are indicated as the error bars in the figure (n = 6).

*GUS* transient expression percentage. Higher *GUS* transient expression was observed under low light intensity (50, 100 $\mu$mol$\cdot$m$^{-2}\cdot$s$^{-1}$) compared to high light intensity (150, 190 $\mu$mol$\cdot$m$^{-2}\cdot$s$^{-1}$), indicating further investigation was needed to understand transformation efficiency

**Table 3. Transformation efficiencies under glufosinate selection after various light intensities during co-cultivation.** Values are present as mean ± standard error (n = 6). Data was analyzed by ANOVA analysis with significant p-value equal or less than 0.05.

| Light Source | Light Intensity ($\mu$mol$\cdot$m$^{-2}\cdot$s$^{-1}$) | Transformation Efficiency [ns] (%) |
|---|---|---|
| White LED | 50 | 4.2 ± 4.6 |
| | 100 | 2.8 ± 2.6 |
| | 150 | 6.9 ± 4.7 |
| | 190 | 5.1 ± 3.9 |
| Fluorescent | 100 | 3.5 ± 3.4 |

[ns]: No statistical difference among the treatments. The p-value of ANOVA analysis for treatment effect is 0.28.

(Shi et al., in press). Very few studies have characterized the impact of light intensity during co-cultivation on the *Agrobacterium*-mediated transformation efficiency. Previous research focused on the comparison between darkness and light environments during co-cultivation [12, 13]. Both studies showed enhanced *GUS* transient expression under light environments in *Arabidopsis thaliana* and *Phaseolus acutifolius* [12, 13]. There are similarities in the promotion of transformation efficient by light between these two previous studies and the present study of HPPD-inhibitor selection. However, our study included a wide range of light intensities (50, 100, 150, 190 μmol·m$^{-2}$·s$^{-1}$) during co-cultivation. Transient expression only reports on the efficiency of transgene delivery into plant cells without selection pressure of integrated T-DNA fragments. In contrast, transformation efficiency is dependent of transgene integration in the plant genome and successful regeneration of resistant cells [24]. Furthermore, the present study also compared white LED and fluorescent light at the equivalent light intensities; these results provide some reference for adapting LED light sources to plant transformation protocols and potential light quality effects on transformation which is further discussed below.

The detection of an effect of light on transformation efficiency exclusively under HPPD selection may be partially attributed to the differential transient expression of each transgene during co-cultivation before the presence of selectable agents. Prior to integration into the host genomic DNA, the T-DNA needs to be imported into the plant nucleus, where the integrated transgene can be expressed. Though no clear timeline of T-DNA transport into the nucleus has been presented, a study demonstrated that *GUS* activity was detected after five days of co-cultivation in soybean when expressed under the control of the *35S* promoter [11]. During the co-cultivation stage of *Agrobacterium*-mediated transformation in soybean, transgene expression is possible. Therefore, expression-associated factors may contribute to the differences in transformation efficiency observed between the two selective markers. Firstly, the use of different promoters to drive expression of two selective markers, PcsvmvXYYZ for *hppdPf-4Pa* and P35S3 for *bar*, could potentially contribute to the observed variations between them. Promoter activity can be influenced by light intensity and light quality. A previous study has shown that light intensity regulates promoter activity depending on the core promoter architecture [25]. Moreover, the sequence structure of the Pcsvmv core promoter differs from P35S [26], which could result in different levels of expression at varying light intensities. The effects of light intensity have been investigated using transient expression via agroinfiltration. *GUS* transient expression driven by 35S promoter showed no significant difference when leaves were placed under low and high light intensities [27–29]. This result shares similarity with the outcome of the present study under glufosinate selection, where *bar* (selective marker of glufosinate) was driven by P35S promoter. However, since the transformation methods and plant materials differ between the two studies, comparisons should be made with caution. Furthermore, evidence of different responses of promoters to light intensity has been observed in other plant systems. For instance, in grapevine (*Vitis vinifera*), the *VviLAR1* promoter activity increased with rising light intensity, while the *VviLAR2* promoter remained insensitive to various light intensities [30]. Additionally, light quality, such as far-red light, has demonstrated regulation on a promoter activity in photoautotrophic cyanobacteria (*Chlorogloeopsis fritschii*) [31]. According to those findings, it is possible that the use of two different promoters led to the different transformation efficiency in the present study.

Differences in the activity of the enzymes encoded by each transgene could also be responsible for the differences in transformation efficiencies under varying light intensities. *hppdPf-4Pa* encodes HPPD-4, a bacterial enzyme that alleviates the inactivation of endogenous HPPD enzymes by HPPD inhibitors in plants [32]. The HPPD protein catalyzes the formation of the precursors of α-tocopherol and plastoquinone which both mitigate adverse high light effect on

plant cells. [32–34]. In addition to its role in electron transfer during photosynthesis, plasto-quinone is essential for biosynthesizing carotenoid that protects plants from radical damage from high light [35]. HPPD-4, as one form of HPPD proteins, possesses the photoprotective function. Increasing light intensity could possibly upregulate the expression of *hppdPf-4Pa* in plant cells during co-cultivation. Under high light intensity at the end of the co-cultivation stage, increased levels of *hppdPf-4Pa* expression could be beneficial for shoot regeneration of transformed cells under HPPD inhibitor selection. The observed increase in transformation efficiency followed a linear regression, prompting the question of whether further increasing light intensity during co-cultivation could lead to a continued improvement of transformation efficiency under HPPD inhibitor selection. The highest light intensity used in the present study, 190 $\mu mol \cdot m^{-2} \cdot s^{-1}$, is higher than the reported light intensity used in previous soybean transformations [14, 15]. However, it is important to recognize that 190 $\mu mol \cdot m^{-2} \cdot s^{-1}$ is not the light saturation point or the excessive light level for most plant growth. High light intensity is usually investigated at levels surpassing 900 $\mu mol \cdot m^{-2} \cdot s^{-1}$ [36]. Consequently, further investigations into the effects of light intensity higher than 190 $\mu mol \cdot m^{-2} \cdot s^{-1}$ on transformation efficiency are needed. *bar* is associated with nitrogen metabolism in plants [19]. In contrast, the expression of *bar* may have little effect on dissipating energy from high light intensity and provides no advantage.

Under HPPD inhibitor selection, the transformation efficiency exhibited numerical differences (not statistically significant) between the co-cultivation under fluorescent light and white LED light at 100 $\mu mol \cdot m^{-2} \cdot s^{-1}$. The main distinction in the spectrum between white LED and fluorescent light used in the present study is the presence of UV light from fluorescent light (Table 1). The measurable UV spectrum (300–400 nm) consists of UV-B (280–320 nm) and UV-A (320–400 nm) both of which can influence plant cell activity [37]. The UV-A has a similar effect on plants as blue light; however, the UV-B can cause various damages to the plant cells due to its high energy and induction of Reactive Oxidative Species (ROS) accumulation [38]. HPPD-4 expressed transiently by *hppdPf-4Pa* in the cell might mitigate the additional energy generated by UV-B of fluorescent light. Furthermore, the potential upregulation of *hppdPf-4Pa* by high ROS during co-cultivation facilitate plant cells against HPPD inhibitor selection. Furthermore, the improvement observed with fluorescent light might be explained by the effect of UV light on *Agrobacterium* infectivity. A previous study showed that the UV light enhanced the infectivity of *A. tumefaciens* despite the inhibition on bacterial viability [39]. Though, the enhancement was not observed under glufosinate selection. Another possible explanation for the difference between two selective agents is that their promoter activities could be affected differently by UV light, as explained in the previous discussion. Therefore, despite the lack of statistical significance in the comparison between fluorescent light and white LED in the current study, the existing evidence justifies further investigation into the influence of spectral variations on transformation efficiency. This experiment has shed light on the influence of light intensity and spectrum on transformation efficiency, offering prospects for future optimization of transformation protocols based on environmental conditions. Nevertheless, further investigations are warranted to provide additional evidence and a deeper understanding of the observed outcomes. Specifically, exploring the effects of higher photon flux densities, UV-B light, and considering different promoter options could enhance our comprehension of the factors influencing transformation efficiency in soybean and other plant species.

## Supporting information

**S1 File.**
(DOCX)

## Acknowledgments

The authors express their gratitude to BASF Agricultural Solutions for funding this research (grant number 4-20-B00454-3). Special thanks are extended to Dr. Alexander Galle and Dr. Jixiang Kong from BASF Agricultural Solutions for their scientific input and guidance on the potential applications of this research. The authors also wish to acknowledge Dr. Wusheng Liu and Dr. Kedong Da from NCSU for their expert advice on tissue culture and molecular biology techniques.

## Author Contributions

**Conceptualization:** Xiaonan Shi, Baochun Li, Ricardo Hernández.

**Data curation:** Xiaonan Shi.

**Formal analysis:** Xiaonan Shi, Ricardo Hernández.

**Funding acquisition:** Baochun Li, Ricardo Hernández.

**Investigation:** Xiaonan Shi.

**Methodology:** Xiaonan Shi, Baochun Li, Marcela Rojas-Pierce, Ricardo Hernández.

**Project administration:** Xiaonan Shi, Ricardo Hernández.

**Resources:** Xiaonan Shi, Ricardo Hernández.

**Software:** Xiaonan Shi.

**Supervision:** Ricardo Hernández.

**Validation:** Xiaonan Shi, Ricardo Hernández.

**Visualization:** Xiaonan Shi, Ricardo Hernández.

**Writing – original draft:** Xiaonan Shi.

**Writing – review & editing:** Xiaonan Shi, Baochun Li, Marcela Rojas-Pierce, Ricardo Hernández.

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
