## [Decision Letter · Decision Letter 0]

10 Jul 2024

PONE-D-24-24813White LED Intensities During Co-Cultivation Affect the Agrobacterium-mediated Soybean (Glycine max) Transformation Using Mature Half Seeds as ExplantsPLOS ONE

Dear Dr. Hernández,

Thank you for submitting your manuscript to PLOS ONE. After careful consideration, we feel that it has merit but does not fully meet PLOS ONE’s publication criteria as it currently stands. Therefore, we invite you to submit a revised version of the manuscript that addresses the points raised during the review process.

We look forward to receiving your revised manuscript.

Kind regards,

Marcos Pileggi, Ph.D

Academic Editor

PLOS ONE

Journal Requirements:

"BASF Corporation provided partial funding."

4. Thank you for stating the following in the Financial Disclosure section: 

""BASF Corporation provided partial funding."

We note that you received funding from a commercial source: BASF Corporation

Within this Competing Interests Statement, please confirm that this does not alter your adherence to all PLOS ONE policies on sharing data and materials by including the following statement: "This does not alter our adherence to PLOS ONE policies on sharing data and materials.” (as detailed online in our guide for authors http://journals.plos.org/plosone/s/competing-interests).  

If there are restrictions on sharing of data and/or materials, please state these. Please note that we cannot proceed with consideration of your article until this information has been declared. 

5. We note that your Data Availability Statement is currently as follows: 

"All relevant data are within the manuscript and its Supporting Information files."

**Additional Editor Comments:**

The authors  need  to  address   all the  points  highlighted  by the  reviewers to advance  in the manuscript evaluation process.

Reviewers' comments:

Reviewer's Responses to Questions

**Comments to the Author**

1. Is the manuscript technically sound, and do the data support the conclusions?

Reviewer #1: Yes

Reviewer #2: Yes

2. Has the statistical analysis been performed appropriately and rigorously? 

Reviewer #1: Yes

Reviewer #2: Yes

3. Have the authors made all data underlying the findings in their manuscript fully available?

Reviewer #1: Yes

Reviewer #2: Yes

4. Is the manuscript presented in an intelligible fashion and written in standard English?

Reviewer #1: Yes

Reviewer #2: Yes

5. Review Comments to the Author

Reviewer #1: The present manuscript highlights the role of white LED light during co-cultivation to improve transformation efficiency. Even though the work appears to be fascinating, the manuscript requires answers to various questions (added as comments to authors as major and minor) to be scientifically sound. Hence, I accept this manuscript for major revision.

Major comments:

1. The abstract could be improved by highlighting the key results. The authors should also mention the transformation efficiency achieved using this method in the abstract.

2. The introduction lacks novelty and Authors could include any report showing positive effects of LED in improving transformation efficiency in any crop.

3. How affordable is the HPPD inhibitor Tembotrione? Is this inhibitor inexpensive enough to be used as a selection agent to improve transformation efficiency in soybean?

4. Have the authors checked the minimum inhibitory concentration (MIC) using glufosinate in the soybean variety Throne? If so, please include this data as supplementary information.

5. In the Materials and Methods section, line 116, it is mentioned as Agrobacterium tumefaciens strain EHA105 carrying the plasmid construct. Please specify the name of the plasmid.

6. In line 180, the authors mention co-cultivation for 5 days. How did the authors arrive at this conclusion? Did they conduct previous studies to determine that 5 days of co-cultivation is optimal, or was this duration chosen arbitrarily?

7. In line 183, the authors used three antibiotics (timentin, 200 mg/L cefotaxime, 50 mg/L vancomycin). Did these antibiotics not hinder the growth of the transgenic plant?

8. Statistical analysis is missing in Tables 2 and 3.

9. The clarity of the photos in Figures 3 and 4 should be improved (around 600 dpi).

10. The authors mention stable transformation. Have they conducted segregation pattern analysis for T1 plants? If so, please include this data.

Minor comments

11. The overall language of the manuscript should be improved.

12. The authors should also check for spelling mistakes, degrees, and various subscripts mentioned in the manuscript to ensure they are correctly depicted.

Reviewer #2: Authors present interesting findings on the effect of white LED intensities on frequency of Agrobacterium mediated genetic transformation in soybean. The evaluation was done at co-cultivation stage, which is the most critical phase of transformation, wherein bacterial cells carrying the transgene come into contact with target plant tissue for genome alteration to take place. No major issues were found in this manuscript except the ones I provide below:

Manuscript Title and Abstract are fine, although can be improved.

Introduction section can be improved. Some context is required where Authors stated that “From which half-seed explants derive, mature seeds are easy to store and obviate the need for greenhouse facilities”. It is not clear how/why those conditions would be required for explant preparation and tissue culture, since the steps carried out are done under in vitro culture conditions.

Furthermore, not only half-seed method facilitates shoot organogenesis, but that can also be achieved using other forms of explants as well, including exposure of cells to Agrobacterium.

For Results and subsequent sections

Glufosinate selection- additional information must be provided on how transformation efficiency under this selection was determined. How results were recorded and calculated should be included in the MM.

Authors should perhaps provide result on shoot induction, elongation and rooting frequencies to also give a clear picture of the effect of LED intensities on regeneration, which is also a critical part of recovering transgenic plants. Such revelations will also give an idea of the physiological status of the tissue totipotency after LED treatments. Others remaining sections of the manuscript are sufficient.

6. PLOS authors have the option to publish the peer review history of their article (what does this mean?). If published, this will include your full peer review and any attached files.

Reviewer #1: **Yes: **Muthukrishnan Arun

Reviewer #2: No

---

## [Author Response · Author response to Decision Letter 0]

31 Aug 2024

Response to Reviewers

Reviewer #1: The present manuscript highlights the role of white LED light during co-cultivation to improve transformation efficiency. Even though the work appears to be fascinating, the manuscript requires answers to various questions (added as comments to authors as major and minor) to be scientifically sound. Hence, I accept this manuscript for major revision.

Major comments:

1. The abstract could be improved by highlighting the key results. The authors should also mention the transformation efficiency achieved using this method in the abstract.

The results were highlighted in line 25 -26 and line 30- 33. The transformation efficiency achieved is mentioned as a range in line 23 - 25.

2. The introduction lacks novelty and Authors could include any report showing positive effects of LED in improving transformation efficiency in any crop.

The novelty of this study has been re-emphasized at the end of each paragraph in the introduction section, as shown in line 48-50, line 77-84, line103-105.

As for the LED effects on transformation efficiency, after an extensive literature search, no publication has been found demonstrating such effects in any plant species. Authors reiterated the absence of this kind study in line 74-77.

3. How affordable is the HPPD inhibitor Tembotrione? Is this inhibitor inexpensive enough to be used as a selection agent to improve transformation efficiency in soybean?

The HPPD inhibitor Tembotrione is relatively affordable. The tembotrione price listed on a chemical science company in the United States is about $100 per 100 mg. Moreover, the effective concentration of tembotrione used in the selection is much lower (0.066 mg/L in this study) compared to glufosinate (6 mg/L). 

4. Have the authors checked the minimum inhibitory concentration (MIC) using glufosinate in the soybean variety Throne? If so, please include this data as supplementary information.

Thank you for suggesting this. Yes. The inhibitory effects were tested at 0, 2, 4, 6, 8 mg/L prior to transformation experiments. Data is included in Table S1 of the supplemental materials document.

5. In the Materials and Methods section, line 116, it is mentioned as Agrobacterium tumefaciens strain EHA105 carrying the plasmid construct. Please specify the name of the plasmid.

The plasmid is provided by BASF and used internally only. The plasmid is labeled as “pBNCL2020”. The information is included in line 134.

6. In line 180, the authors mention co-cultivation for 5 days. How did the authors arrive at this conclusion? Did they conduct previous studies to determine that 5 days of co-cultivation is optimal, or was this duration chosen arbitrarily?

According to most of the publications regarding soybean Agrobacterium-mediated transformation, the co-cultivation usually takes 5 days. The authors used this as the reference to determine 5-day period for co-cultivation. This is emphasized in line 177-179.

7. In line 183, the authors used three antibiotics (timentin, 200 mg/L cefotaxime, 50 mg/L vancomycin). Did these antibiotics not hinder the growth of the transgenic plant?

No additional tests were performed to evaluate the antibiotic effects on explant growth. The present study implemented common transformation practice as the baseline, therefore, the rates of antibiotics used are referred to published literatures. One previous study suggested negative effect of high concentration of cefotaxime and vancomycin (higher than 500 mg/L) on soybean tissue after extensive period of times (longer than 63 days) (Wiebke et al., 2006). Additionally, the antibiotics were eliminated starting from rooting stage on in the present study. Based on those findings, the authors hypothesize that the antibiotics used in the present study have little or minimal hindering effects on transgenic plant. 

Reference:

Wiebke, Beatriz & Ferreira, Fabricio & Pasquali, Giancarlo & Bodanese-Zanettini, Maria & Droste, Annette. (2006). Influence of antibiotics on embryogenic tissue and Agrobacterium tumefaciens suppression in soybean genetic transformation. Bragantia. 65. 10.1590/S0006-87052006000400002.

8. Statistical analysis is missing in Tables 2 and 3.

No statistical analysis was conducted for environmental data in Table 2, The mean of all the values collected were presented, we included the standard deviation. This reporting format is recommended by: 

 “Committee on Controlled Environment Technology and Use” (https://www.controlledenvironments.org/guidelines-for-measuring-and-reporting-environmental-parameters-in-growth-chambers/) 

The statistical analysis method was added in the caption for Table 3 in line 283-284. The non-significant result was indicated by superscript “ns” in the footnote.

9. The clarity of the photos in Figures 3 and 4 should be improved (around 600 dpi).

The dpi of Figures 3 and 4 have been adjusted to 600 dpi.

10. The authors mention stable transformation. Have they conducted segregation pattern analysis for T1 plants? If so, please include this data.

Due to time and financial constraints, the authors were not able to carry T0 plants to progeny. 

Minor comments:

11. The overall language of the manuscript should be improved.

Language improvement was primarily conducted in discussion section, showing as sentences in line 318-320, 330-333, 377-379, 383-386.

12. The authors should also check for spelling mistakes, degrees, and various subscripts mentioned in the manuscript to ensure they are correctly depicted.

The manuscript has been checked for spelling, format, and grammars. 

Reviewer #2: Authors present interesting findings on the effect of white LED intensities on frequency of Agrobacterium mediated genetic transformation in soybean. The evaluation was done at co-cultivation stage, which is the most critical phase of transformation, wherein bacterial cells carrying the transgene come into contact with target plant tissue for genome alteration to take place. No major issues were found in this manuscript except the ones I provide below:

Manuscript Title and Abstract are fine, although can be improved.

Abstract was improved by highlighting the results and stating the study’s significance in line 23 to 27 and line 31 to 34.

Introduction section can be improved. Some context is required where Authors stated that “From which half-seed explants derive, mature seeds are easy to store and obviate the need for greenhouse facilities”. It is not clear how/why those conditions would be required for explant preparation and tissue culture, since the steps carried out are done under in vitro culture conditions.

Thank you for pointing it out. Changes were made in line 60-64 as “Various explant types have been employed in Agrobacterium-mediated transformation systems, including cotyledonary node from immature seed (Olhoft et al.), hypocotyl from germinated seedlings (Wang & Xu, 2008), and half-seed from mature seed (Luth et al., 2015). The utilization of half-seed explants has proven advantageous. Obtaining immature seeds and seedlings requires continuous plant care and maintenance in greenhouses or growth chambers for extended periods before the transformation process. From which half-seed explants derive, mature seeds are easy to store and obviate the need for greenhouse facilities”.

The advantage of half-seed is reflected on the easy pre-transformation explant preparation. The immature seed requires soybean plants at reproductive stage, which takes significant amount of time and greenhouse space. The hypocotyl explants are derived from days-old seedlings that takes additional efforts to obtain the materials to begin transformation process.

Furthermore, not only half-seed method facilitates shoot organogenesis, but that can also be achieved using other forms of explants as well, including exposure of cells to Agrobacterium.

The argument was clarified by changing the sentence to “Additionally, the half-seed method enables rapid and direct shoot organogenesis from the meristem, bypassing the callogenesis stage. This is crucial for expediting the regeneration process of genetically modified shoots” in line 66-68. The authors’ intention here is to state the rapid shoot organogenesis achieved by using meristem.

For Results and subsequent sections

Glufosinate selection- additional information must be provided on how transformation efficiency under this selection was determined. How results were recorded and calculated should be included in the MM.

Sentence was added in line 275-277, stating “The transformation efficiency was determined by using equation (1) after confirming transformation event by PCR analysis on rooted soybean plants”.

The determination of transformation efficiency is described in the added subsection “Transformation Efficiency Determination” under MM in line 250-254.

Authors should perhaps provide result on shoot induction, elongation and rooting frequencies to also give a clear picture of the effect of LED intensities on regeneration, which is also a critical part of recovering transgenic plants. Such revelations will also give an idea of the physiological status of the tissue totipotency after LED treatments. Others remaining sections of the manuscript are sufficient.

Thank you very much for this insightful comment. For the current experiment, no data was collected on the rates of shoot induction, elongation, and rooting since it was not part of our stated hypotheses. However, we have conducted a separate experiment examining the impact of different light intensities on the induction and elongation stages. The results of this study will be published as a separate journal article.

---

## [Decision Letter · Decision Letter 1]

1 Oct 2024

White LED Intensities During Co-Cultivation Affect the Agrobacterium-mediated Soybean (Glycine max) Transformation Using Mature Half Seeds as Explants

PONE-D-24-24813R1

Dear Dr. Hernández,

We’re pleased to inform you that your manuscript has been judged scientifically suitable for publication and will be formally accepted for publication once it meets all outstanding technical requirements.

Kind regards,

Marcos Pileggi, Ph.D

Academic Editor

PLOS ONE

Additional Editor Comments (optional):

Reviewers' comments:

Reviewer's Responses to Questions

**Comments to the Author**

1. If the authors have adequately addressed your comments raised in a previous round of review and you feel that this manuscript is now acceptable for publication, you may indicate that here to bypass the “Comments to the Author” section, enter your conflict of interest statement in the “Confidential to Editor” section, and submit your "Accept" recommendation.

Reviewer #1: All comments have been addressed

2. Is the manuscript technically sound, and do the data support the conclusions?

Reviewer #1: Yes

3. Has the statistical analysis been performed appropriately and rigorously? 

Reviewer #1: Yes

4. Have the authors made all data underlying the findings in their manuscript fully available?

Reviewer #1: Yes

5. Is the manuscript presented in an intelligible fashion and written in standard English?

Reviewer #1: Yes

6. Review Comments to the Author

Reviewer #1: (No Response)

7. PLOS authors have the option to publish the peer review history of their article (what does this mean?). If published, this will include your full peer review and any attached files.

Reviewer #1: **Yes: **Muthukrishnan Arun

---

## [Editor Report · Acceptance letter]

15 Nov 2024

PONE-D-24-24813R1 

PLOS ONE

Dear Dr. Hernández, 

I'm pleased to inform you that your manuscript has been deemed suitable for publication in PLOS ONE. Congratulations! Your manuscript is now being handed over to our production team.

Kind regards, 

on behalf of

Dr. Marcos Pileggi 

Academic Editor

PLOS ONE